# The Use of Real-Time PCR for the Pathogen Quantification in Breeding Winter Wheat Varieties Resistant to Eyespot

**DOI:** 10.3390/plants11111495

**Published:** 2022-06-02

**Authors:** Jana Palicová, Pavel Matušinsky, Veronika Dumalasová, Alena Hanzalová, Ivana Svačinová, Jana Chrpová

**Affiliations:** 1Crop Research Institute, Drnovská 507, 161 06 Prague, Czech Republic; dumalasova@vurv.cz (V.D.); hanzalova@vurv.cz (A.H.); chrpova@vurv.cz (J.C.); 2Agrotest Fyto, Ltd., Havlíčkova 2787/121, 767 01 Kroměříž, Czech Republic; matusinsky@vukrom.cz (P.M.); svacinova@vukrom.cz (I.S.); 3Department of Botany, Faculty of Science, Palacký University in Olomouc, Šlechtitelů 27, 783 71 Olomouc, Czech Republic

**Keywords:** *Oculimacula yallundae*, *Oculimacula acuformis*, *Pch1* gene, molecular markers, qPCR

## Abstract

The reaction of twenty-five winter wheat cultivars frequently grown in the Czech Republic to inoculation with *Oculimacula yallundae* and *Oculimacula acuformis* was evaluated in small plot trials from 2019 to 2021. The eyespot infection assessment was carried out visually using symptoms on stem bases and quantitative real-time polymerase chain reaction (qPCR). The cultivars were also tested for the presence of the resistance gene *Pch1* using the STS marker *Xorw1*. Statistical differences were found between cultivars and between years. The lowest mean level of eyespot infection (2019–2021) was visually observed in cultivar Annie, which possessed resistance gene *Pch1*, and in cultivar Julie. Cultivars Turandot and RGT Sacramento were the most susceptible to eyespot. The method qPCR was able to distinguish two eyespot pathogens. *O. yallundae* was detected in higher concentrations in inoculated plants compared with *O. acuformis*. The relationship between the eyespot symptoms and the pathogen’s DNA content in plant tissues followed a moderate linear regression only in 2021. The highest eyespot infection rate was in 2020 due to weather conditions suitable for the development of the disease.

## 1. Introduction

The main pathogens of stem-base diseases of cereals belong to the following genera of fungi: *Oculimacula*, *Rhizoctonia*, *Fusarium*, *Microdochium* and *Gaeumannomyces.* Yield losses due to these diseases can reach up to 40%. Eyespot is the most serious disease in this group. It is caused by two different species: *Oculimacula yallundae* (Wallwork and Spooner) Crous and W. Gams and *Oculimacula acuformis* (Nirenberg) Y. Marín and Crous, which have similar life-cycles [1]. On the other hand, both fungi differ in morphology, pathogenicity, occurrence and sensitivity to fungicides [2,3]. Eyespot pathogens have a wide host range among cereals and grasses. *Oculimacula yallundae* (OY) was prevalent in winter wheat samples infected by eyespot in the Czech Republic [4], whereas *O. acuformis* (OA) was predominant in winter rye eyespot samples in Lithuania [5]. *Oculimacula* spp. are supposed to survive on plant debris for more than 3 years and their occurrence varies due to weather conditions [6,7].

Conidia produced on infected straw are the principal form of inoculum in the field [8]. The conidia are dispersed over short distances in rain splash droplets and can initiate infections on wheat from autumn to spring. The pathogens penetrate leaf sheaths up to the stem. The first symptoms in the form of non-specific necrosis on leaf sheaths can be visible at the growth stage of leaf development (BBCH 13–14). *Oculimacula yallundae* generally grows more rapidly through leaf sheaths than *O. acufomis* [9]. The colonization by *O. acuformis* increases later in the season. Later, at the growth stage of tillering and stem elongation, typical elliptical eye-shaped spots with diffuse margins can be seen. The increase in lesion extent appeared to stop approximately at the stage BBCH 70–71 [9].

Visual assessment of symptoms on infected wheat stem bases cannot discriminate between OY and OA, and their presence can be masked by the less damaging pathogens of stem base disease, especially early in the growing season [10,11]. A real-time polymerase chain reaction assay, suitable for large scale testing, was developed and used for quantitative detection and discrimination of OY and OA [12].

There are three characterized sources of eyespot resistance used in commercial wheat varieties: the dominant resistant gene *Pch1* from *Aegilops ventricosa* [13], *Pch2* from *Triticum aestivum* (cultivar Cappelle Desprez) [14] and a quantitative trait locus (QTL) *Q.Pch.jic-5A* [15]. The most effective and also the most widely used resistance gene in commercial cultivars is *Pch1*, a single major gene mapped to the distal end of the long arm of chromosome 7D.

The objective of this study was to evaluate the resistance of selected winter wheat cultivars to eyespot in a small plot infection experiment using quantitative real-time PCR (qPCR) and visual assessment. The effect of the *Pch1* eyespot resistance gene in the tested cultivars was evaluated.

## 2. Results

### 2.1. Visual Assessment of Eyespot Symptoms on Wheat Cultivars and Detection of Pch1 Gene in Tested Cultivars 

Statistically significant differences in eyespot infection were found between tested winter wheat cultivars in years 2019–2021 using ANOVA (Table 1). The highest level of visible eyespot symptoms was in 2020 due to weather conditions suitable for the development of the disease. The lowest level of visible eyespot symptoms was detected in 2021. Only two cultivars possessed the *Pch1* gene of resistance to eyespot: Annie and Illusion. Cultivar Annie had the lowest level of eyespot symptoms from the tested collection. A low level of infection was also observed in cultivar Julie. The majority of the cultivars were moderately resistant to moderately susceptible, and the differences were rather small (see Table 2). Cultivars LG Orlice, Frisky, LG Mocca, Illusion, RGT Cesario, Genius, Asory and Balitus showed a mean level of infection up to 3.0. The most infected cultivars by eyespot were Turandot and RGT Sacramento, which had a susceptible reaction.

### 2.2. Real-Time PCR Evaluation of the Pathogen Content in Tested Wheat Cultivars 

*Oculimacula* spp. quantification based on qPCR analysis showed significant differences among tested cultivars in the case of *O. yallundae* during the years 2019–2021 (Figure 1). The most resistant cultivar Annie (carrying the gene of resistance to eyespot *Pch1*) was set as a control, and the relative amount of *O. yallundae* DNA in other cultivars measured by qPCR was related to this control sample as fold difference (FD). The lowest DNA level was determined in cultivar Annie (*Pch1*, FD 1) and in cultivars KWS Elementary (FD 2.21), Turandot (FD 3.07), Asory (FD 3.31), Collector (FD 3.88), Pirueta (FD 4.43), Sally (FD 4.62), Illusion (*Pch1*, FD 4.67) and Johnson (FD 4.68). The highest level of DNA was detected in cultivars Gaudio (FD 20.43) and Steffi (FD 17.06). 

*Oculimacula acuformis* was detected with much lower intensity than *O. yallundae* and differences among the cultivars were not statistically significant (Figure 2). The lowest amount of *O. acuformis* DNA was detected in cultivars Collector (FD 0.34), Chevignon (FD 0.43), Butterfly (FD 0.44) and Gaudio (0.44). The highest DNA level was found in cultivars Steffi (FD 1.15) and Fakir (FD 1.00), which had the same DNA level as the control cultivar, Annie. The cultivar Collector was among the least infected with *O. yallundae* and *O. acuformis* by qPCR assessment.

The relationship between the eyespot symptoms and the pathogen’s DNA content in plant tissues followed a moderate linear regression only in 2021 (R^2^ = 0.3897, *p* = 0.001).

## 3. Discussion

The severity of eyespot infection assessed visually and by the molecular method qPCR was dependent on weather patterns each year. Although conditions were constant in the small plot experiment, the level of infection varied from year to year. The presence of the gene of resistance to eyespot *Pch1* in cultivars Annie and Illusion was found to be important in the context of eyespot infection. The cultivar Annie was the least infected cultivar in the tested set on average 2019–2021. The cultivar Illusion was the sixth least infected by eyespot in an average of three years. The maximum visual assessment of cultivar Illusion was 3.6 in 2020 when the weather conditions were the most favorable for eyespot infection development. In general, cultivars possessing gene *Pch1* have been characterized by high resistance to eyespot in Germany [16]. In the case of cultivar Illusion, further observations are needed in climatically different years.

*Oculimacula acuformis* was detected at a very low level in the small plot experiments. It seems *O. acuformis* did not play an important role even in a trial plot infection, although the amount of *O. yallundae* and *O. acuformis* inoculum was comparable. *Oculimacula acuformis* is predominantly a pathogen of rye, and results of the present study showed that the impact to this pathogen in wheat samples from 2019 to 2021 was very low. *Oculimacula yallundae* was more aggressive to the wheat samples. This is consistent with previous reports [17]. *Oculimacula yallundae* might grow more rapidly through leaf sheaths than *O. acuformis* at the beginning of infection [9]. *Oculimacula yallundae* tends to cause earlier stem lesions and colonization, while *O. acuformis* infection increases later in the season [18]. OA could infect and grow more rapidly through leaf sheaths during cold winters [19], while under less cold conditions, OY could grow through the leaf sheaths more rapidly and colonize the stem first, so by reaching the stem earlier, the symptoms of OY are more advanced [20]. Although the assessment of the small plot experiment was performed at the stage BBCH 73–77 and the increase in lesion extent appeared to stop at BBCH 70–71 [9], a year with a prevalence of OA was not observed.

The linear regression analysis indicates that the relation of visual assessment and relative quantity obtained from qPCR showed a low correlation coefficient, which means a weak relationship between the symptoms and pathogen DNA content in plant tissues. In a previous study [17], the relationship between the eyespot symptoms and the pathogen DNA content in plant tissues followed a moderate linear regression. In the current study, qPCR results of *O. yallundae* DNA content in plant tissues and visual symptoms followed a moderate linear regression only in 2021. Real-time PCR can be a useful supporting method for testing a large amount of new breeding lines or cultivars to eyespot resistance. This long breeding process usually ends with the registration of a new variety. According to our results, the qPCR method can be applied to eyespot diagnostic assays, including wheat cultivar resistance assessment. Real-time PCR is very sensitive and can distinguish small differences among tested materials, even in years that are unfavorable to disease development. Moreover, the level of *O. yallundae* and *O. acuformis* can be easily checked by qPCR and it is possible to monitor the occurrence of both species in the field. This is very important for the control strategy because both pathogens differ in morphology, pathogenicity, occurrence and sensitivity to fungicides [2]. However, it is always necessary to supplement the molecular results with a visual assessment.

## 4. Materials and Methods

### 4.1. Visual Assessment of Eyespot Symptoms on Wheat Cultivars and Detection of Pch1 Gene in Tested Cultivars

The reaction of 25 selected winter wheat cultivars to inoculation with *Oculimacula yallundae* and *O. acuformis* was studied in a small plot trial in Prague-Ruzyně (50.0864797 N, 14.3020897 E) in three farming seasons: 2018/2019, 2019/2020 and 2020/2021. All tested cultivars were registered in the Czech Republic by the Central Institute for Supervising and Testing in Agriculture. The resistant control was cultivar Annie, possessing gene *Pch1* [21].

The inoculum for the small plot trial was prepared from a mixture of 12 isolates of OY and 7 isolates of OA. These isolates were obtained from different locations of the Czech Republic. Fungi were cultivated on PDA (Himedia) at 20 °C in the dark for 14 days. Mycelium with agar was cut into 5 × 5 mm squares and each Erlenmeyer flask containing sterilized barley grains was inoculated with 4 squares of one *Oculimacula* isolate. The barley grains (50 g/250 mL Erlenmeyer flask) were sterilized three times for 20 min at 120 °C before inoculation. The barley grains inoculated with *Oculimacula* spp. were incubated under UV light at 18 °C for about 4–5 weeks. After this time, the inoculum was removed from the flasks, mixed in a large container and directly applied on experimental plots in December and in March (40 g/m^2^). The reaction of tested cultivars was rated at the milk growth stage (BBCH 73–77). A 0 to 5 rating scale was used (0—no symptoms; 1—one small spot; 2—more spots covering maximally half of the stem perimeter; 3—spots covering more than half of the stem perimeter; 4—spot covering the whole stem perimeter; 5—broken stem). In inoculated plots, 60 randomly selected stems were assessed. 

The winter wheat cultivars from the experimental years 2019–2021 were screened with STS marker *Xorw1* [22] to identify the presence or absence of *Pch1* gene, as described by Dumalasová et al. [21].

### 4.2. Real-Time PCR Evaluation of the Pathogen Content in Tested Wheat Cultivars 

Samples of the lower parts of the stem base (approximately 20 mm long segments) were collected annually in the milk-wax growth stage (end of June). Stem bases (20 pcs) of each cultivar were thoroughly dried, homogenized to a fine powder and subsequently processed according to the methodology published in previous work by Palicová et al. [17]. All tested winter wheat cultivars were evaluated by quantitative real-time PCR (qPCR) by using primers: Oculimacula-R (universal reverse), Ac F-D (specific to OA) and Yall F-H (specific to OY), and reference gene phenylalanine ammonia-lyase (primers WpalF/R) previously published by Walsh et al. [12].

### 4.3. Statistical Analysis

Each experiment was set up in randomized repeats and results were expressed as mean ± standard error (SE). The data were analyzed by the statistical software Statistica 14.0.0.15 (Statsoft Inc., Tulsa, OK, USA). A general factorial ANOVA (Analysis of Variance) was performed at 95% confidence interval and 5% level of significance. When the *p*-value was less than 0.05, the Fisher’s Least Significant Difference (LSD) test for multiple comparisons was carried out. The significantly different mean values were represented by different letters. The relationship between visual eyespot symptoms and pathogens’ DNA content in plant tissue was studied using a Regression Analysis. 

## 5. Conclusions

Real-time PCR proved to be a sensitive method for eyespot pathogen quantification in winter wheat varieties. It can help in a long-term breeding process, where thousands of accessions need to be assessed and only the best ones selected. It is important to observe the incidence of both causal agents of eyespot in the field, mainly because of their different sensitivity to fungicides. According to the results of this study, *Oculimacula yallundae* was more aggressive than *O. acuformis* in wheat under the studied conditions. It was confirmed that the gene of resistance to eyespot *Pch1* plays a significant role in decreasing eyespot symptoms on stem bases in wheat cultivars.

## Figures and Tables

**Figure 1 plants-11-01495-f001:**
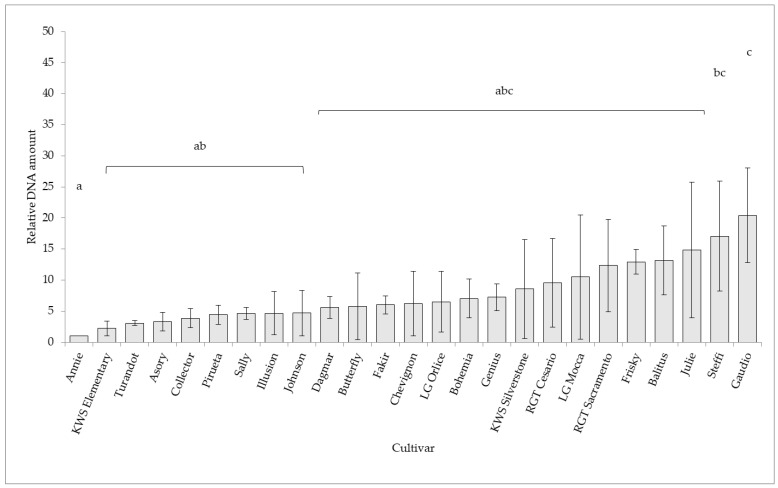
The qPCR assessment of *Oculimacula yallundae* in the winter wheat cultivars (2019–2021). In the ANOVA with a multiple comparison Fisher´s LSD test (*p* < 0.05), homogeneous groups are marked with the same letters.

**Figure 2 plants-11-01495-f002:**
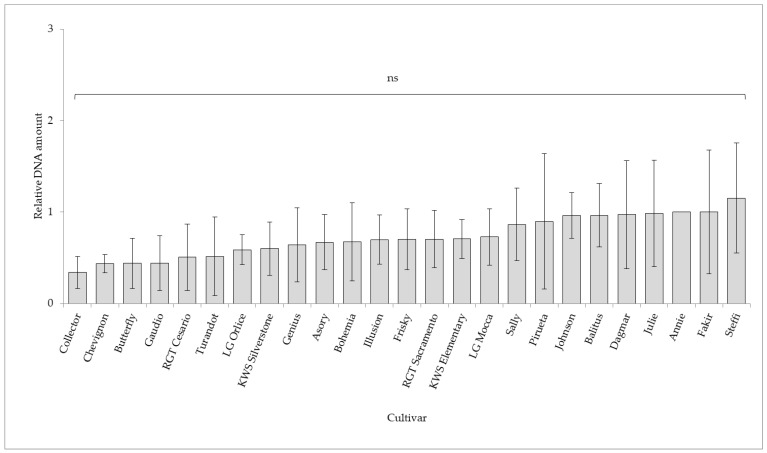
The qPCR assessment of *Oculimacula acuformis* in the winter wheat cultivars (2019–2021). The ANOVA was not statistically significant (ns).

**Table 1 plants-11-01495-t001:** The ANOVA results for the visual assessment of the wheat cultivars inoculated by eyespot (2019–2021).

Effect	SS	Df	MS	*F*-Ratio	*p*-Value
Intercept	40,036.859	1	40,036.859	34,180.563	0.000
CULTIVAR	460.611	24	19.192	16.385	0.000
YEAR	1098.175	2	549.088	468.771	0.000
CULTIVAR*YEAR	469.266	48	9.776	8.346	0.000
Error	5172.611	4416	1.171		

SS—sum of squares; Df—degrees of freedom; MS—mean square.

**Table 2 plants-11-01495-t002:** Evaluation of eyespot symptoms on winter wheat cultivars after *Oculimacula* spp. inoculation in three-year trials in Prague-Ruzyně (2019–2021).

Cultivar	*Pch1*	2019	2020	2021	Mean	H. g.
Annie	+	2.1	2.2	1.1	2.0	a
Julie	-	3.2	2.3	1.4	2.3	ab
LG Orlice	-	3.2	2.2	2.7	2.7	bc
Frisky	-	3.2	2.9	2.2	2.8	cd
LG Mocca	-	2.9	3.0	2.7	2.9	cde
Illusion	+	2.8	3.6	2.3	2.9	cde
RGT Cesario	-	3.4	3.7	1.8	3.0	cde
Genius	-	3.2	3.7	2.0	3.0	cde
Asory	-	2.7	4.1	2.2	3.0	cde
Balitus	-	3.2	3.7	2.2	3.0	cdef
Bohemia	-	3.4	3.6	2.2	3.1	cdef
Collector	-	3.0	3.8	2.7	3.1	cdef
Pirueta	-	3.8	3.7	1.8	3.1	cdef
Fakir	-	3.8	3.3	2.1	3.1	cdefg
Dagmar	-	3.6	3.8	2.0	3.1	cdefg
Johnson	-	3.3	3.5	2.7	3.2	defg
Steffi	-	3.6	4.0	2.0	3.2	defg
KWS Silverstone	-	3.3	3.4	2.9	3.2	defg
Sally	-	3.4	3.7	2.5	3.2	efg
Gaudio	-	3.9	3.5	2.3	3.2	efg
Chevignon	-	3.4	3.8	2.4	3.2	efg
Butterfly	-	3.4	3.7	2.8	3.3	efg
KWS Elementary	-	3.4	3.6	3.3	3.4	fg
RGT Sacramento	-	3.5	3.8	3.2	3.5	g
Turandot	-	3.5	4.1	2.4	3.5	g

Scale 0–5 (0–2 resistant, 3 moderately resistant to moderately susceptible, 4–5 susceptible); H. g.—Homogeneous groups (the significantly different mean values were represented by different letters a–g; *p* = 0.05, Fisher’s LSD test).

## Data Availability

The data can be provided by the authors.

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
