# Peer review of "The Use of Real-Time PCR for the Pathogen Quantification in Breeding Winter Wheat Varieties Resistant to Eyespot"

_plants, 2022, doi:10.3390/plants11111495_

Round 1

Reviewer 1 Report

Results

There is question regarding terminology, Authors speak about “resistance”, but I do not think, that results show true resistance. For example, authors have determined that ‘Steffi’ was susceptible in 2020 (evaluation = 4) and resistant in 2021 (evaluation = 2). Is this cultivar susceptible or resistant? In my opinion in this case it is severity of disease depending on cultivars.

I do not agree with Line 78, severity of diseases was not lower for cultivar ‘Illusion’ to compare with others cultivars.

I do not agree with Line 136 – 137. Cultivar ‘Illusion’ was not the most resistant in accordance with data in the Table 1.

What was the idea of Figure 3? Of course, amount of fungus is significantly higher in the end of season, it is not new idea. Scattering of amount of fungus between samples was high, but there is no attempt to discuss it.

Correlation between amount of Oculimacula spp. (determined by Real-Time PCR) and severity of symptoms was low. For example, level of fungus was low in cultivar ‘Turandot’, but symptoms were clearly expressed – mean value 3.5. How Real-Time PCR could be used for evaluation of new breeding lines?

I do not think, that analyse of 13 samples from one field, in one year is not enough to make conclusions about prevalence of Oculimacula species.

Methods and Materials

Methods were not described in details. How did you make mixture of fungal isolates? How many mycelia was used to inoculate grains?

How did you collect samples in the field?

Author Response

Dear Reviewer

Thank very much for all comments. We have accepted all suggestions for changes and tried to implement them as best we could. Below is a detailed description of our responses to the suggestions.

There is question regarding terminology, Authors speak about “resistance”, but I do not think, that results show true resistance. For example, authors have determined that ‘Steffi’ was susceptible in 2020 (evaluation = 4) and resistant in 2021 (evaluation = 2). Is this cultivar susceptible or resistant? In my opinion in this case it is severity of disease depending on cultivars.

Response: Thank you for the terminology correction, we agree. We tried to avoid the term “resistance” and replaced it with “level of infection, symptoms and severity of disease” (lines 69, 76, 87, 125, 126, 164).

I do not agree with Line 78, severity of diseases was not lower for cultivar ‘Illusion’ to compare with others cultivars.

Response: We agree with the reviewer, this part was removed.

I do not agree with Line 136 – 137. Cultivar ‘Illusion’ was not the most resistant in accordance with data in the Table 1.

Response: We agree with the reviewer, this part was more discussed in the text (lines 126–131).

What was the idea of Figure 3? Of course, amount of fungus is significantly higher in the end of season, it is not new idea. Scattering of amount of fungus between samples was high, but there is no attempt to discuss it.

Response: We agree with the reviewer’s opinion and we removed Figure 3 from the article. We also removed from the text of the article the discussion highlighted by the reviewer.

Correlation between amount of Oculimacula spp. (determined by Real-Time PCR) and severity of symptoms was low. For example, level of fungus was low in cultivar ‘Turandot’, but symptoms were clearly expressed – mean value 3.5. How Real-Time PCR could be used for evaluation of new breeding lines?

Response: We thank the reviewer for the very helpful comment. We agree with the reviewer's opinion and we have changed our statement. In view of the correlations found in the current study, we propose this method not as a stand-alone method but only as a "supporting method" and we added this new statement form in the new version of the paper. “Real-time PCR can be a useful supporting method for testing a big amount of new breeding lines or cultivars to eyespot resistance.” and “But it is always necessary to supplement the molecular results with a visual assessment.”

I do not think, that analyse of 13 samples from one field, in one year is not enough to make conclusions about prevalence of Oculimacula species.

 Response: We agree with the reviewer’s opinion and we removed Figure 3 from the article. We also removed from the text of the article the discussion highlighted by the reviewer.

Methods and Materials

Methods were not described in details. How did you make mixture of fungal isolates? How many mycelia was used to inoculate grains?

Response: The methods were described in more details (lines 173–182).

How did you collect samples in the field?

Response: We agree with the reviewer’s opinion and we removed the parts concerning the field evaluation from the new version of the article.

Sincerely,

Jana Palicová et al.

Reviewer 2 Report

In this original research article the Authors applied a Real-Time PCR diagnostic method to detect and quantify the infection of eyespot disease caused by the fungi Oculimacula yallundae and O. acuformis in winter wheat varieties selected in the Czech Republic and compared the results of molecular diagnosis with those obtained using the visual assessment of the disease symptoms. The diagostic method proved to be useful for both breeding programs and epidemiological investigations.

The experimental design is well conceived and the objectives clearly defined. The methods are adequate but more details must be added to their description. The results are clearly presented and discussed properly. 

I made only minor text edits and added few recommendations such as the suggestion to describe the method of articial inoculation of wheat cultivars more in detail (see notes in the text, attached file)

Author Response

Dear Reviewer,

Thank you very much for all comments. We have accepted all suggestions for changes and tried to implement them as best we could. Below is a detailed description of our responses to the suggestions.

I made only minor text edits and added few recommendations such as the suggestion to describe the method of artificial inoculation of wheat cultivars more in detail (see notes in the text, attached file).

Response:

We corrected minor text edits.

The method of inoculation was described in more details (lines 173–182).

Sincerely,

Jana Palicová et al.

Round 2

Reviewer 1 Report

I think, you have improved text significantly and I hope manuscript will be published.

Author Response

Thank you very much for your comments and suggestions.